# Characteristics and Risk Factors of Ultra-High-Risk Patients with Newly Diagnosed Multiple Myeloma

**DOI:** 10.3390/jpm13040666

**Published:** 2023-04-14

**Authors:** Chuanying Geng, Guangzhong Yang, Huixing Zhou, Huijuan Wang, Yanchen Li, Yun Leng, Zhiyao Zhang, Yuan Jian, Wenming Chen

**Affiliations:** Department of Hematology, Beijing Chaoyang Hospital, Capital Medical University, Beijing 100020, China

**Keywords:** multiple myeloma, ultra-high-risk, survival, prognosis

## Abstract

Objective: To investigate the clinical characteristics and risk factors of ultra-high-risk (UHR) patients with newly diagnosed multiple myeloma (MM). Methods: We screened UHR patients with a survival of less than 24 months and we selected patients with a concurrent survival of more than 24 months as a control group. We retrospectively analyzed the clinical characteristics of UHR patients with newly diagnosed MM and screened related risk factors. Results: In total we analyzed 477 patients, which included 121 (25.4%) UHR patients and 356 (74.6%) control patients. Median overall survival (OS) and progression-free survival (PFS) of UHR patients was 10.5 months (7.5–13.5 months) and 6.3 months (5.4–7.2 months), respectively. Univariate logistic regression analysis showed that age > 65 years, hemoglobin (HGB) < 100 g/L, lactate dehydrogenase (LDH) > 250 U/L, serum creatinine (SCr) > 2 mg/dL, corrected serum calcium (CsCa) > 2.75 mmol/L, B-type natriuretic peptide (BNP) or N-terminal prohormone BNP (NT-proBNP) > 2 upper limit of normal (ULN), high-risk cytogenetics, Barthel index score, and International Staging System (ISS) stage III were associated with UHR MM. In a multivariate analysis, age > 65 years, LDH > 250 U/L, CsCa > 2.75 mmol/L, BNP or NT-proBNP > 2 ULN, high-risk cytogenetics, and Barthel index score were independent risk factors for UHR MM. Moreover, UHR patients had a worse response rate than control patients. Conclusion: Our study highlighted the characteristics of UHR MM patients and suggested that the combination of organ insufficiency and highly malignant myeloma cells resulted in poor outcomes of patients with UHR MM.

## 1. Introduction

Multiple myeloma (MM) is a hematological malignancy characterized by an abnormal proliferation of neoplastic plasma cells in the bone marrow [1]. New treatments have greatly improved the survival of patients, but the survival of MM patients is heterogeneous [2]. Accurate assessment of the prognosis for patients is an important issue in the diagnosis and treatment of MM [3]. Ultra-high risk (UHR) patients have significantly shorter survival; however, common staging systems cannot effectively distinguish UHR patients [4]. In addition, there is no uniform definition for survival among UHR patients. Some studies have defined survival of less than 24 months as UHR MM, while others have defined survival of less than 12 months as UHR patients [5,6]. Screening for UHR in patients with MM and individualized treatments are of great significance to improve the prognosis for MM patients. In this study, the disease characteristics and risk factors for UHR MM patients with a survival time of less than 24 months were investigated.

We screened newly diagnosed MM patients at Beijing Chaoyang Hospital, Capital Medical University, and analyzed the clinical characteristics and screened the prognostic factors of UHR MM patients.

## 2. Methods

### 2.1. Patients

Baseline data of newly diagnosed MM patients in Beijing Chaoyang Hospital Affiliated to Capital Medical University from 1 January 2010 to 31 December 2021 were collected. Patients were diagnosed according to the IMWG definition of MM and were followed until 1 August 2022 [7]. Patients’ data were retrospectively collected and followed up through the electronic medical record System (EMRS). Baseline data included sex, age, MM subtype, ISS stage, hemoglobin (reference range 120–170 g/L), serum creatinine (SCr, reference range 0.5–1.0 mg/dL), corrected serum calcium (CsCa, reference range 2.11–2.52 mmol/L), lactate dehydrogenase (LDH, reference range 120–250 U/L), B-type natriuretic peptide (BNP, reference range 0–100 pg/mL), N-terminal prohormone BNP (NT-proBNP, reference range 0–228 pg/mL), cytogenetic abnormalities (CAs), Barthel index score, and induction therapy regimens. Barthel index scores were calculated before induction therapy. Barthel index consists of 10 items: feeding, bathing, grooming, dressing, defecation, bladder control, using the toilet, chair transfer, walking and climbing stairs. Each item was graded according to the amount of assistance required to complete each activity. All baseline data are routine testing items for newly diagnosed MM patients at our center. Cytogenetic abnormalities included del (17p), t (14; 16) and t (4; 14) were detected by fluorescence in situ hybridization (FISH) in myeloma cells purified by anti-CD138 + magnetic beads. The study was approved by the Medical Ethics Committee of Beijing Chaoyang Hospital in accordance with the Declaration of Helsinki.

### 2.2. Response and Outcome Measures

Patients were evaluated according to IMWG criteria, including strict complete response (sCR), complete response (CR), very good partial response (VGPR), partial response (PR), stable disease (SD), and progressive disease (PD) [8]. The primary end points of this study were progression-free survival (PFS) and overall survival (OS). PFS was defined as the time from patient diagnosis to disease progression or death, and OS was defined as the time from patient diagnosis to death from any cause or the date of last follow-up. Patients who could not be followed up were censored on the date of their last follow-up.

### 2.3. Statistical Analysis

All data in this study were analyzed by SPSS 23.0 software. The t test and chi-square test were used to analyze the differences between the two groups. Survival curves were generated using the Kaplan–Meier method, and the Log rank test was used to evaluate the difference between survival curves. Univariate logistic regression and multivariate logistic regression analysis were used to determine the risk factors for UHR MM. Statistical measurement of risk was reported as the odds ratio (OR) and 95% confidence interval (CI). Results were considered statistically significant when *p* < 0.05.

## 3. Results

### 3.1. Patient Characteristics

Of the 477 patients enrolled in this study, 25.4% (121) had a survival time of less than 24 months (Table 1). The median follow-up time for all patients was 36.7 (range 0.2–117.3) months. Median OS and PFS of UHR patients were 10.5 months (7.5–13.5 months) and 6.3 months (5.4–7.2 months), respectively (Figure 1). The median age of the patients was 60 years (30–87 years), and the male to female ratio was 1.13 (253/224). The largest proportion of patients were IgG type MM patients (49.2%), and half of the patients were in ISS stage III. All patients received induction regimens containing at least one new drug (bortezomib, thalidomide, lenalidomide); 292 patients (61.2%) received bortezomib-based regimen, 46 patients (9.6%) received a regimen of immunomodulatory agents (IMIDs), and 139 patients (29.1%) received bortezomib combined with IMIDs. After induction therapy, 142 (29.8%) patients received autologous stem cell transplant (ASCT). Patients with UHR MM had fewer opportunities for ASCT compared to control patients. As shown in Table 1, there were statistically significant differences between UHR patients and control patients in gender, age, ISS stage, HGB, SCr, CsCa, LDH, BNP, NT-proBNP, high-risk CAs, and Barthel index scores. Compared to the control group, UHR patients were more elderly and more UHR patients were in ISS III stage and had high-risk CAs. The HGB level and Barthel index score of UHR MM patients were lower, while the SCr, CsCa, LDH, BNP, and NT-proBNP levels were higher. There was no difference in the type of MM or induction regimens.

### 3.2. Response

Evaluation of the best response showed that 407 patients (85.3%) achieved PR or better after treatment (Table 2). One hundred and thirty patients (27.3%) achieved sCR, 59 (12.4%) CR, 100 (21.0%) achieved VGPR, and 118 (24.7%) achieved PR. The overall response rate (ORR) was significantly lower in UHR patients than in controls (59.5% vs. 94.1%, *p* < 0.001). Deep responses were also different, with VGPR rates of 12.4% versus 23.9%, CR rates of 8.3% versus 13.8%, and sCR rates of 5.0% versus 34.8% in UHR and control groups, respectively. Fewer UHR patients achieved VGPR or better compared to controls (25.6% vs. 72.5%, *p* < 0.001).

### 3.3. Risk of UHR Patients

Univariate logistic regression analysis showed nine factors associated with UHR MM: age > 65 years, HGB < 100 g/L, LDH > 250 U/L, SCr > 2 mg/dl, CsCa > 2.75 mmol/L, BNP or NT-proBNP > 2 ULN, Barthel index score, high risk CAs, and ISS stage (Table 3). Multivariate analysis was performed using these nine covariates. Six factors were independently associated with UHR in the multivariate analysis; these included age > 65 years (OR = 2.026, *p* = 0.007), LDH >250 U/L (OR = 2.117, *p* = 0.026), CsCa > 2.75 mmol/L (OR = 2.406, *p* = 0.015), BNP or NT-proBNP > 2 ULN (OR = 2.198, *p* = 0.007), high risk CAs (OR = 2.141, *p* = 0.007), and Barthel index score (OR = 0.983, *p* = 0.004) (Table 3).

## 4. Discussion

With the use of new drugs, the survival of multiple myeloma patients has greatly improved, with some young patients surviving approximately 10 years [2]. However, MM is still an incurable disease, and for some UHR patients, remission is difficult to achieve through induction therapy, or they have early recurrence after remission, resulting in significantly shorter survival. Currently, there is no uniform definition of UHR patients’ survival [9,10,11]. In this study, UHR MM patients with a survival of less than 24 months were screened to investigate their clinical features and risk factors.

There are many prognostic factors associated with MM. Currently, most studies have confirmed that high serum β2-microglobulin, low albumin, high serum creatinine, high lactate dehydrogenase, low hemoglobin, low platelet count, high-risk cytogenetic abnormalities, and advanced age are adverse prognostic factors for multiple myeloma [3]. The ISS is a commonly used risk stratification system for newly diagnosed MM patients. It stratifies MM patients according to serum β2-microglobulin and serum albumin levels to identify high-risk MM patients [4]. Subsequent studies confirmed that certain cytogenetic abnormalities including del (17p), t (4; 14), and t (14; 16) were associated with a poor prognosis for MM patients. In 2015, IMWG proposed the revised ISS (R-ISS) staging system, which combines high risk cytogenetics, serum LDH, and ISS [12]. R-ISS is a simple but clinically useful system for predicting the OS and PFS of newly diagnosed MM patients. Cytogenetic abnormalities are important prognostic factors for MM. There are only three cytogenetic abnormalities in the R-ISS. Other abnormalities such as t (14; 20), 1q21 amplification, and 1p deletion are also associated with a poor prognosis and may be markers of high-risk diseases [9]. In this study, univariate analysis suggested that ISS stage III and high-risk cytogenetic abnormalities including del (17p), t (4; 14), and t (14; 16) were risk factors for UHR. In multivariate analysis, high risk cytological abnormalities remained an independent risk factor for UHR, but ISS stage III was not. This suggested that ISS staging is not an effective tool for identifying UHR and that high-risk cytogenetic abnormalities are still important for identifying UHR.

Previous studies showed age was significantly associated with survival of patients with MM. Ludwig H et al. [13] analyzed 10,549 patients with MM in North America, Europe, and Japan and found that survival decreased steadily by each decade from age 50 to age over 80 years. Moreover, patients with MM who were younger than 50 years of age had longer OS than patients who were more than 50 years old regardless of traditional or high-dose chemotherapy [14]. Older patients often presented with poor prognostic factors and had several comorbidities which limited the selection of treatment. At present, ASCT remains the standard treatment after induction therapy for eligible patients with newly diagnosed MM [15]. Barlogie B et al. [16] randomly enrolled 516 MM patients, 261 of whom received ASCT and 255 of whom received standard dose chemotherapy. The results showed that there was no significant difference in efficacy between the two groups. Long-term follow-up showed that MM patients who received ASCT had significant survival benefits and had longer PFS and OS. Rosenberg A. S et al. [17] analyzed 13,494 patients with newly diagnosed MM from 1998 to 2012 and used propensity scores to balance the characteristics of patients in the ASCT and non-ASCT groups. Among them, a total of 20.8% patients received ASCT after induction therapy, and the number of patients receiving ASCT increased yearly. Age was also a factor for patients receiving ASCT, with 37.6% of patients under 60 years of age receiving ASCT, compared to 11.5% of patients between 60 and 79 years of age. The median time for patients to receive ASCT was 9.4 months after diagnosis, and most patients received ASCT within 2 years after diagnosis. The median OS of patients who received ASCT was 72.9 months. When ASCT was performed, the patient’s survival improved. Among patients who received ASCT, survival was longer among those who received ASCT within 12 months of diagnosis compared to those who received ASCT more than 12 months after diagnosis. In the era of highly effective induction therapy, ASCT could still further improve patient survival. Kumar L et al. [18] studied 349 MM patients who received ASCT between 1995 and 2016. After ASCT, 90.8% of patients obtained PR or better and 61% of patients obtained CR. After a median follow-up of 73 months, patients had a median OS of 90 months and a median PFS of 41 months. This study suggested that patients receiving ASCT have increased CR rates and longer OS compared to standard therapy. Compared to conventional chemotherapy, ASCT after induction therapy could further improve the depth of remission and prolong the PFS and OS of patients with newly diagnosed MM, and has been used as the standard treatment for eligible patients with newly diagnosed MM. Currently, after effective induction therapy, ASCT remains the preferred treatment for patients over 65 years of age with good overall physical status scores. However, due to the high financial burden and low ASCT acceptance, some patients who were eligible for ASCT refused to receive it in China. In our center, although we actively recommended ASCT to patients who are eligible for ASCT, only half of patients aged less than 65 years and few patients over 65 years of age received ASCT. At present, we continue to educate patients in the hope of increasing their acceptance of ASCT and improving their survival. In China, age is still one of the important factors that limits patients from receiving ASCT. We also found that age over 65 years was an independent poor prognostic factor for UHR MM patient. This may be related to the fact that few patients over 65 years of age received ASCT.

Patients with MM may present with hypercalcemia, one of the diagnostic markers of MM. Malignant myeloma cells secrete several cytokines, such as tumor necrosis factor, interleukin-3, receptor activation of NFκB ligand, and macrophage inflammatory protein-1α, which increase osteoclaster-mediated bone destruction. Furthermore, myeloma cells can interact with the bone marrow microenvironment, leading to osteoclast activation and the secretion of osteoblast suppressor factors. Myeloma cells can cause osteolytic damage, which causes calcium ions to flood into the blood, resulting in hypercalcemia [19,20]. Hypercalcemia occurs in about 20% of MM patients. Hypercalcemia was a poor prognostic factor for OS among patients with MM, and was included in the DS stage system [21]. Zagouri F et al. [22] analyzed 2129 MM patients to evaluate the impact of hypercalcemia on prognosis in newly diagnosed MM patients, and found that 19.5% of MM patients had hypercalcemia at the time of diagnosis. This study showed that hypercalcemia was associated with anemia, thrombocytopenia, lower glomerular filtration rate, advanced ISS stage, and osteolytic lesions. Hypercalcemia was more common in high-risk cytogenetic patients and was associated with poor survival. They suggested that hypercalcemia remained a poor prognostic factor for MM patients in the age of new drugs.

In addition to the risk stratification system, genomic characteristics, and tumor load, other immutable patient-related factors influenced the prognosis of MM. Organ insufficiency was a poor prognostic factor for MM. Baseline rates of cardiovascular complications, including hypertension and heart failure, tend to be higher as patients age. Multiple myeloma occurs more frequently in elderly patients and is often accompanied by cardiovascular complications. Heart rate (HR) is strongly associated with cardiovascular morbidity and mortality in a variety of diseases. Wang J et al. [23] evaluated the prognostic potential of HR in patients with MM. They retrospectively analyzed 386 MM patients and found that all-cause mortality was higher in patients with HR >100 beats per minute (bpm) than in patients with 60 ≤ HR ≤ 100 bpm and <60 bpm. BNP is mainly secreted by the ventricle. In heart failure, the secretion of BNP by the ventricle increases significantly, and the degree of its increase is positively correlated with the severity of heart failure, which can be used as an indicator to evaluate the process of heart failure. NT-proBNP is an N-terminal precursor BNP which is also a sensitive indicator for monitoring heart failure. Pavo N et al. [24] conducted a retrospective analysis on 118 MM patients and found that NT-proBNP was positively correlated with β2-microglobulin and MM disease progression [ISS stage 1 133.3 pg/mL, ISS stage 2 487.4 pg/mL, ISS stage 3 969.1 pg/mL]. During follow-up, NT-proBNP levels were significantly elevated along with other markers of MM disease severity in patients. It suggested that elevated levels of NT-proBNP were associated with disease severity of MM patients. BNP and NT-proBNP were associated with heart function, Semochkin SV et al. [25] prospectively analyzed 20 patients with newly diagnosed MM and patients with high levels of NT-proBNP had shorter OS. This study also found that BNP or NT-proBNP > 2 ULN was an independent risk factor for UHR MM.

Clinical frailty and geriatric assessments had been shown to affect the prognosis of MM, but their routine use was largely limited by clinical time constraints [26,27,28]. International Myeloma Working Group (IMWG) analyzed 869 newly diagnosed elderly patients with MM and performed geriatric assessment. Based on age, comorbidity, cognition, and physical condition, IMWG developed a scoring system to classify patients into three groups: healthy group, moderately healthy group and frail group. The 3-year overall survival rate in the healthy group was significantly higher than that in the moderately healthy group and frail group. At 12 months, the cumulative incidence of non-hematologic adverse events ≥ 3 was lower in the healthy group than in the moderately healthy group and frail group. This frailty score could predict mortality and toxicity risk in elderly patients with MM [26]. The basic activities of daily life had a great influence on the prognosis of MM patients, and could be used as a prognostic factor of MM patients. The inability of patients to perform basic activities without assistance was an important indicator for evaluating host factors. Barthel index was created in 1965 to assess functional independence of 10 daily activities and was commonly used in the field of rehabilitation [29]. Barthel index was mainly used for patients with neurological diseases and the elderly [30,31]. Yang H et al. [30] tested the reliability and validity of the improved Barthel index as an evaluation tool for daily living activities of patients with ischemic stroke. They analyzed 231 patients with ischemic stroke and used Rasch analysis to evaluate the reliability and validity of the improved Barthel index. It was found that the reliability of the improved Barthel index was high, but the matching degree between project difficulty and patient ability was poor. Dos Santos Barros V et al. [32] verified the reliability, internal consistency and structural validity of the Barthel index in the palliative care of cancer patients in Brazil. The Barthel Index, Karnofsky Performance Scale (KPS) and European Organization for Research in the Treatment of Cancer Questionnaire-core 15 (EORTC-QLQ-C15-PAL) were used to evaluate 220 cancer patients. It is suggested that Barthel index had sufficient correlation with KPS and EORTC-QLQ-C15-PAL functional capacity domains. Morishima T et al. [33] analyzed newly diagnosed patients with gastric cancer, colorectal cancer or lung cancer from 35 hospitals in Osaka Prefecture, Japan, during 2010–2014. In this study, dysfunction was classified into three levels (none, moderate and severe) based on the Barthel index score. The study suggested that a higher risk of death was significantly associated with both moderate and severe dysfunction. Integrated consideration of functional status in cancer diagnosis may improve the prediction of survival time in young and middle-aged cancer patients. Barthel index was the most widely used tool to evaluate the basic activities of daily life, but few studies evaluated its prognostic value in MM. In this study, both univariate and multivariate analyses suggested that low Barthel index was a prognostic factor for patients with UHR.

The prognostic factors of MM were not only related to the factors of the myeloma cells, but also related to patient factors. In this study, we screened six factors including age > 65 years, LDH > 250 U/L, CsCa > 2.75 mmol/L, BNP or NT-proBNP > 2 ULN, high-risk cytogenetics, and Barthel index score, which were independent risk factors for UHR MM. LDH, CsCa, and high-risk cytogenetics were related to the biological characteristics of myeloma cells. Age, BNP/NT-proBNP, and Barthel index score were related to short survival. Combining the biological characteristics of myeloma cells and patient characteristics was more conducive to the identification of UHR MM.

MM is a malignant hematological disease with very high heterogeneity. UHR patients have more aggressive myeloma cells that require effective treatment to improve patient survival. Patients with UHR MM need to receive multiple therapies to improve organ function; this would provide patients with more effective therapies to control myeloma cell proliferation. Identifying patients with UHR MM and using individualized treatment measures to address the patient’s tolerance and improve efficacy could effectively improve the patient’s prognosis [34,35].

This study has several limitations. It was a retrospective analysis using data from a single MM diagnosis and treatment center; the results need to be confirmed by more research centers. New drugs including bortezomib and IMIDs have significantly improved the survival of patients with MM. The median follow-up time in this study was short; long-term follow-up is required to verify the results. After the initial treatment in our center, some patients were transferred to other centers for further treatment, leading to the loss of follow-up of some patients, which may affect the results of the study.

## 5. Conclusions

In conclusion, UHR MM patients with organ dysfunction were very fragile and were less likely to receive the ASCT. Combination of organ insufficiency and high malignant myeloma cells resulted in worse outcome of patients with UHR MM.

## Figures and Tables

**Figure 1 jpm-13-00666-f001:**
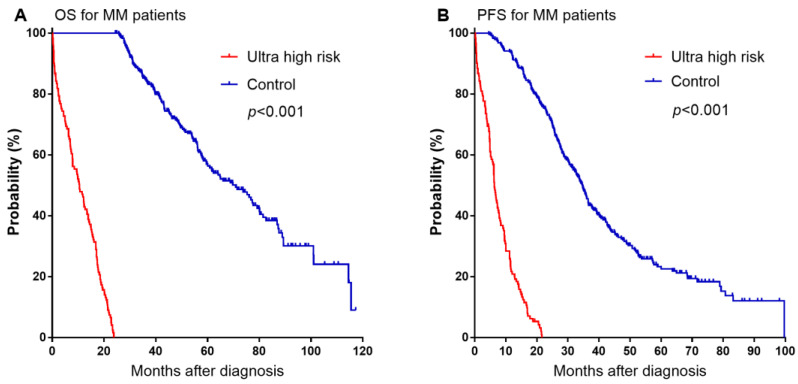
Kaplan-Meier survival curves on OS and PFS among patients with newly diagnosed MM. (**A**) OS. (**B**) PFS.

**Table 1 jpm-13-00666-t001:** Baseline clinical and biological characteristics of MM patients.

	All Patients	Short Survival Patients	Control	
Characteristics	*n* = 477	*n* = 121	*n* = 356	
	*n* (%)	*n* (%)	*n* (%)	*p* value
Sex				
Male	253 (53.0)	74 (61.2)	179 (50.3)	0.038
Female	224 (47.0)	47 (38.8)	177 (49.7)	
Age				
≤65 years	313 (65.6)	65 (53.7)	248 (69.7)	0.001
>65 years	164 (34.4)	56 (46.3)	108 (30.3)	
MM subtype				
IgG	230 (48.2)	61 (50.4)	169 (47.5)	0.676
IgA	103 (21.6)	24 (19.8)	79 (22.2)	
IgD	21 (4.4)	5 (4.1)	16 (4.5)	
Light chain only	111 (23.3)	26 (21.5)	85 (23.9)	
Non-secretory	12 (2.5)	5 (4.1)	7 (2.0)	
ISS stage				
I	63 (13.2)	7 (5.8)	56 (15.7)	0.001
II	171 (35.8)	36 (29.8)	135 (37.9)	
III	243 (50.9)	78 (64.5)	165 (46.3)	
Hemoglobin				
<100 g/L	320 (67.1)	92 (76.0)	228 (64.0)	0.015
≥100 g/L	157 (32.9)	29 (24.0)	128 (36.0)	
Serum creatinine				
≤2 mg/dL	360 (75.5)	74 (61.2)	286 (80.3)	0.000
>2 mg/dL	117 (24.5)	47 (38.8)	70 (19.7)	
Corrected serum calcium				
≤2.75 mmol/L	411 (86.2)	89 (73.6)	322 (90.4)	0.000
>2.75 mmol/L	66 (13.8)	32 (26.4)	34 (9.6)	
Lactate dehydrogenase				
≤250 U/L	405 (84.9)	88 (72.7)	317 (89.0)	0.000
>250 U/L	72 (15.1)	33 (27.3)	39 (11.0)	
BNP or NT-proBNP				
≤2 times ULN	291 (72.4)	57 (52.3)	234 (79.9)	0.000
>2 times ULN	111 (27.6)	52 (47.7)	59 (20.1)	
High risk CAs by FISH				
yes	132 (27.7)	47 (38.8)	85 (23.9)	0.001
no	345 (72.3)	74 (61.2)	271 (76.1)	
Barthel index score	82.6 ± 21.8	72.7 ± 25.4	86.0 ± 19.3	0.000
Induction regimes				
Bortezomib based	292 (61.2)	73 (60.3)	219 (61.5)	0.075
IMiDs based	46 (9.6)	14 (11.6)	32 (9.0)	
Bortezomib and IMiDs based	139 (29.1)	34 (28.1)	105 (29.5)	
ASCT				
yes	142 (29.8)	12 (9.9)	130 (36.5)	0.000
no	335 (70.2)	109 (90.1)	226 (63.5)	

Abbreviations: UHR: Ultra high-risk; ISS: International Staging System; BNP: B-type natriuretic peptide; NT-proBNP: N-terminal prohormone BNP; ULN: upper limit of normal; FISH: fluorescence in situ hybridization; High risk cytogenetic abnormalities (CAs): del (17p13) or t (14;16) or t (4;14); IMiDs: immunomodulatory agents; ASCT: autologous stem cell transplant.

**Table 2 jpm-13-00666-t002:** Best response rate of MM patients.

	All Patients	Short Survival Patients	Control
	*n* = 477	*n* = 121	*n* = 356
	*n* (%)	*n* (%)	*n* (%)
sCR	130 (27.3)	6 (5.0)	124 (34.8)
CR	59 (12.4)	10 (8.3)	49 (13.8)
VGPR	100 (21.0)	15 (12.4)	85 (23.9)
PR	118 (24.7)	41 (33.9)	77 (21.6)
SD	46 (9.6)	26 (21.5)	20 (5.6)
PD	24 (5.0)	23 (19.0)	1 (0.3)

Abbreviations: UHR: Ultra high-risk; sCR, stringent complete response; CR, complete response; VGPR, very good partial response; PR, partial response; SD, stable disease; PD, progressive disease.

**Table 3 jpm-13-00666-t003:** Logistic regression analysis (univariate and multivariate) for short survival patients.

	**Univariate**			**Multivariate**		
	**OR**	**95% CI**	** *p* **	**OR**	**95% CI**	** *p* **
Age > 65 years	1.978	1.296–3.019	0.002	2.062	1.214–3.502	0.007
HGB < 100 g/L	1.781	1.113–2.850	0.016			0.587
LDH > 250 U/L	3.048	1.812–5.128	0.000	2.117	1.092–4.105	0.026
SCr > 2 mg/dL	2.595	1.656–4.067	0.000			0.668
CsCa > 2.75 mmol/L	3.405	1.991–5.824	0.000	2.406	1.189–4.870	0.015
BNP or NT-proBNP > 2 ULN	3.618	2.257–5.801	0.000	2.198	1.243–3.889	0.007
High risk CAs by FISH	2.025	1.305–3.142	0.002	2.141	1.226–3.738	0.007
Barthel index score	0.975	0.965–0.984	0.000	0.983	0.972–0.995	0.004
ISS stage						
I	1.000	Ref	0.001	1.000	Ref	0.846
II	2.133	0.896–5.079	0.087			0.580
III	3.782	1.648–8.678	0.002			0.589
Induction regimes						
Bortezomib based	1.000	Ref	0.704			
IMiDs based	1.312	0.664–2.595	0.434			
Bortezomib and IMiDs based	0.971	0.608–1.552	0.904			

Abbreviations: OR: Odds ratio; HGB: hemoglobin; LDH: Lactate dehydrogenase; SCr: Serum creatinine; CsCa: corrected serum calcium; BNP: B-type natriuretic peptide; NT-proBNP: N-terminal prohormone BNP; ULN: upper limit of normal; FISH: fluorescence in situ hybridization; High risk cytogenetic abnormalities (CAs): del (17p13) or t (14;16) or t (4;14); IMiDs: immunomodulatory agents.

## Data Availability

The authors confirm that the data that support the findings of this study are available from Wenming Chen, upon reasonable request.

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
