# Peer review of "Characteristics and Risk Factors of Ultra-High-Risk Patients with Newly Diagnosed Multiple Myeloma"

_jpm, 2023, doi:10.3390/jpm13040666_

Round 1

Reviewer 1 Report (New Reviewer)

Author Response

Thank you very much for your letter and advice on our manuscript. We have resubmited new version of manuscript accordance with recommendations and hope that the revision is acceptable and look forward to hearing from you soon.

  1. The article is interesting, as it looks at the very hard to treat patients with multiple myeloma with short survival times

Response: No reply required.

  1. It would be appropriate to provide normal ranges for hemoglobin, lactate dehydrogenase , serum creatinine, corrected serum calcium either in the introduction, or elsewhere in the article.

Response: It was described in the manuscript.

  1. You state that newly diagnosed patients were screened at single hospital. Did you select all newly diagnosed multiple myeloma patients in the said time frame? Were there any bias ?

Response: Some patients were not tested for BNP or NT-proBNP prior to initial treatment and were not included in this study. Because FISH test is not covered by medical insurance, some patients did not receive FISH test, and these patients were not included in this study. The control group of this study was patients with a survival time of more than 24 months, and some patients with a follow-up time of less than 24 months were not included in this study. We screened patients with complete data, excluded only patients with incomplete data and survivors with follow-up time less than 24 months. There were not any bias.

  1. It is interesting to note a significant sex difference in the UHR group, with males outnumbering females by 50%. (table 1).It is unclear why. Were there other risk factors for men that contribute to shorter survival? Was there a competing risk analysis done?

Response: In this study, the OS of female patients was 56.0 months and that of male patients was 48.3 months, but there was no statistical difference between them (p=0.053). We did not further analyze the effect of gender on survival in myeloma patients. Overall life expectancy in China is 77.4 years, but there is a significant gender difference, with 80.5 years for women and 74.7 years for men, a difference of nearly six years. Differences in the general health status of Chinese men and women may lead to differences in OS among male and female patients.

  1. While you have included Barthel Index score at baseline that indicates the functional status of patients, a commonly used co-morbidity index like Charlson co-morbidity index that includes conditions like diabetes mellitus, CHF, other cancers, liver, lung and heart diseases has not been used/described.

Response: The questions raised by the reviewers were very constructive. Barthel Index score is a routine score for inpatients, but Charlson co-morbidity index is not a routine score item in our center. Since this study is a retrospective analysis, there are no baseline data on patients with Charlson co-morbidity index, so no analysis was conducted.

  1. There were only about 30% patients in both groups who received both, IMid and Bortezomib based therapies. (table 1) Is this because of lack of access to the IMId?

Response: Lenalidomide was included in Medicare five years ago, but only one is covered when combined with bortezomib in newly diagnosed myeloma patients. In this study, most patients who were enrolled after 2010 could only receive thalidomide, but the combination of thalidomide and bortezomib resulted in severe peripheral neuropathy, so the proportion of patients with bortezomib combined with IMIDs was not high.

  1. And a majority of patients in the “control’ group did not receive ASCT (only 36.5% patients did). What was the reason for this?

Response: The low percentage of patients receiving transplants is mainly due to the high financial burden and the low acceptance of ASCT by patients. Transplant-related drugs are not covered by the medical insurance, and the patient's financial burden is too heavy to bear the ASCT related requirements. The patient was afraid and worried about ASCT and refused transplantation. We have been educating our patients in the hope of increasing their acceptance of ASCT.

  1. Under the limitations of the study, you describe that the median follow up time was short (line 198). However, there is no description of the median follow –up time else where in the article.

Response: It was described in the manuscript.

  1. You state that the ‘new treatment significantly improved the survival of the patients (line 197). Which treatments and which group of patients are they referring to?

Response: What we mean is that new drugs, including bortezomib and immunomodulatory agents, have improved survival in myeloma patients. It was described in the manuscript.

Abstract: The abstract has been presented well. The objectives, results, and conclusions are summarized well. However, the authors conclude that “the combination of organ insufficiency and high malignant myeloma cells” resulted in shorter outcomes of patients with UHR MM. (lines 24, 25). It is not clear what the authors imply by high malignant myeloma cells, as it is not elaborated in the body of the article. Also, they use the term ‘shorter outcomes’ (line 24). A more appropriate term would be ‘worse outcomes’.

Response: It was elaborated in the manuscript. We changed the term ‘shorter outcomes’ to ‘worse outcomes’.

For parameters hemoglobin, lactate dehydrogenase , serum creatinine, corrected serum calcium, the normal range for the lab needs to be provided.

Response: It was described in the manuscript.

Introduction: The introduction is well summarized. The authors state that newly diagnosed patients were screened at single hospital. Did they select all newly diagnosed patients?

Response: We answered the question in the previous paragraph.

Methods: Patient selection, data collected are described well. Response and outcome measures and statistical analysis have been described well.

Results: It is interesting to note a significant sex difference in the UHR group, with males outnumbering females by 50%. (table 1)

It is unclear why. Were there other risk factors for men that contributed to shorter survival? While the authors have included Barthel Index score, they have not looked at commonly used co-morbidity indices like Charlson co-morbidity index that includes numerous co-morbid conditions like diabetes mellitus, CHF, other cancers, liver, lung and heart diseases.

Response: We answered the question in the previous paragraph.

There were only about 30% patients in both groups who received both, IMid and Bortezomib based therapies. (table 1)

Is this because of lack of access to the IMId?

Response: We answered the question in the previous paragraph.

And a majority of patients in the “control’ group did not receive ASCT (only 36.5% patients did). What was the reason for this?

Response: We answered the question in the previous paragraph.

3.2 Response: Univariate and multi-variate analysis for short survival patients has been discussed in detail

  1. Discussion

Under discussion, the authors have repeated the UHR myeloma definition (lines 129-133) again, after defining it in the Introduction section (lines 35-39)

Response: We deleted the repeated content.

The authors discuss the study in detail, and offer explanation for their findings.

Under the limitations of the study, the authors describe that the median follow up time was short (line 198). They do not describe the median follow –up time else where in the article.

They state that the ‘new treatment significantly improved the survival of the patients (line 197). Which treatments and which group of patients are they referring to?

Response: We answered the question in the previous paragraph.

Reviewer 2 Report (New Reviewer)

This is a single center retrospective analysis of patients with myeloma treated in the era which span over more than a decade (which would include some patients treated with effective induction agents towards the latter years), with observations of poorer survival including overall survival of higher risk myeloma patients. 

Questions that remained unanswered are what can be improved by been able to identify this subgroup of patients- alternative induction regimen or other intervention? Is there difference observed based on year of diagnosis? The authors also reported a very low uptake of ASCT in particular for those with ultra HR disease- can some comments be made as to the reasons behind this- is this related to poorer performance status, older age, organ impairment? The authors reported fewer opportunities for ASCT for this group but reasons for this are not provided. It is also surprising that all patients have cytogenetics performed- an amazing achievement in the real world.

Response rate of the UHR patients is interesting- is the poorer response related to Progression rather than just failure to achieve a deeper response compared to controls? Is there any suggestion as to why this patient group fare pooer-?related to delays in treatment, organ impairment, older age

There is an error made in the correlation of a higher HGB (>100) as one of the nine factors associated with UHR myeloma- should this not be HGB <100?  (Section 3.3 and in abstract- contradicts table 3).

Overall, this paper provides observations that are relevant, but improvements can be made by addressing the above points.

Author Response

Thank you very much for your letter and advice on our manuscript. We have resubmited new version of manuscript accordance with recommendations and hope that the revision is acceptable and look forward to hearing from you soon.

Questions that remained unanswered are what can be improved by been able to identify this subgroup of patients- alternative induction regimen or other intervention?

Is there difference observed based on year of diagnosis? The authors also reported a very low uptake of ASCT in particular for those with ultra HR disease- can some comments be made as to the reasons behind this- is this related to poorer performance status, older age, organ impairment? The authors reported fewer opportunities for ASCT for this group but reasons for this are not provided.

Response: There was no difference on years of diagnosis between the two groups. We explained why the proportion of UHR patients receiving transplantation was low and suggested the importance of improving organ function in the manuscript.

It is also surprising that all patients have cytogenetics performed- an amazing achievement in the real world.

Response: Some patients were not tested for BNP or NT-proBNP prior to initial treatment and were not included in this study. Because FISH test is not covered by medical insurance, some patients did not receive FISH test, and these patients were not included in this study. We screened patients with complete data, excluded only patients with incomplete data and survivors with follow-up time less than 24 months.

Response rate of the UHR patients is interesting- is the poorer response related to Progression rather than just failure to achieve a deeper response compared to controls? Is there any suggestion as to why this patient group fare pooer-?related to delays in treatment, organ impairment, older age

Response: Poor response in UHR patients may result in patients being more likely to progress. The reason for the poor prognosis of UHR patients may be related to both myeloma cells and patient factors. It was elaborated in the manuscript.

There is an error made in the correlation of a higher HGB (>100) as one of the nine factors associated with UHR myeloma- should this not be HGB <100?  (Section 3.3 and in abstract- contradicts table 3).

Response: Yes. It was changed in the manuscript.

Overall, this paper provides observations that are relevant, but improvements can be made by addressing the above points.

Round 2

Reviewer 2 Report (New Reviewer)

This is an improved manuscript but some of the questions remained incompletely addressed- in particular the lower uptake of ASCT in patients with ultra HR disease- the authors stated that few patients  aged >65 can receive ASCT in China and that this is an independent poor prognostic factor. Is the exclusion of older patients from ASCT a contributor to their poorer prognosis (ie by not utilizing an effective treatment modality?) The authors have added in significant details on the benefits of ASCT with previous studies but I am uncertain if this added value to the manuscript. 

In addition, the provision of details on the negative impact of tachycardia (Higher HR) is somewhat difficult to follow, discussion on this and BNP/NT-BNP, can be improved to make it easier for the reader to understand this section.

Similarly, the discussion on barthel index, whilst important, can be improved as it is also difficult for the reader to understand and follow. There are lots of details provided,  it would be much more useful to be succinct and summarise the observations and conclusions rather than provision of all the details.

Author Response

Thank you for your very constructive comments on our manuscript. We have resubmited new version of manuscript accordance with recommendations and look forward to hearing from you soon.

This is an improved manuscript but some of the questions remained incompletely addressed- in particular the lower uptake of ASCT in patients with ultra HR disease- the authors stated that few patients  aged >65 can receive ASCT in China and that this is an independent poor prognostic factor. Is the exclusion of older patients from ASCT a contributor to their poorer prognosis (ie by not utilizing an effective treatment modality?) The authors have added in significant details on the benefits of ASCT with previous studies but I am uncertain if this added value to the manuscript.

Response: Compared with conventional chemotherapy, ASCT after induction therapy could further improve the depth of remission and prolong the PFS and OS of patients with newly diagnosed MM, and had been used as the standard treatment for eligible patients with newly diagnosed MM. Currently, after effective induction therapy, ASCT remains the preferred treatment for patients over 65 years of age with good overall physical status scores. However, due to the high financial burden and low ASCT acceptance, some patients who were eligible for ASCT refused to receive it [Transplant-related drugs are not covered by the medical insurance, and the patient's financial burden is too heavy to bear the ASCT related requirements. The patient was afraid and worried about ASCT and refused transplantation.]. In our center, although we actively recommended ASCT to patients who are eligible for ASCT, only half of patients aged less than 65 years received ASCT, and few patients aged over 65 years received ASCT. At present, we continue to educate patients in the hope of increasing their acceptance of ASCT and improving their survival. In China, age is still one of the important factors limiting patients to receive ASCT.

It was described in the manuscript.

In addition, the provision of details on the negative impact of tachycardia (Higher HR) is somewhat difficult to follow, discussion on this and BNP/NT-BNP, can be improved to make it easier for the reader to understand this section.

Response: We changed the discussion in the manuscript.

Similarly, the discussion on barthel index, whilst important, can be improved as it is also difficult for the reader to understand and follow. There are lots of details provided,  it would be much more useful to be succinct and summarise the observations and conclusions rather than provision of all the details.

Response: We summarize the conclusions and changed the discussion in the manuscript.

Round 3

Reviewer 2 Report (New Reviewer)

Much improved version of  this manuscript with discussion points. My concerns have been addressed. Thank you

This manuscript is a resubmission of an earlier submission. The following is a list of the peer review reports and author responses from that submission.

Round 1

Reviewer 1 Report

Authors reviewed a large cohort of patients trying to identify ultra-high risk MM patients. The definition of ultra-high risk is no uniform in literature.

Comments:

1. In methods describe better inclusion criteria and a definition of ultra-high risk. Although described in the introduction it should be included in methods.

2. Please review figure 1. It seems that some patients died and progressed already at diagnosis. Is this correct?

3.Were all baseline tests available for all 477 patients? 

4. Please confirm if cytogenetics was available for all patients.

5. Why authors used ISS and not R-ISS? 

Reviewer 2 Report

I have read the manuscript by Geng et al. with interest, as high-risk multiple myeloma patients are an unmet problem in managing the disease. However, the general impression of the manuscript, despite the potentially important and exciting topic and the large population of the study, is that manuscript is poorly written, with extremely little novelty, and extensive editing of the English language and style is required.

My key findings:

1.      What % of both populations- UHR and controls underwent autologous hematopoietic stem cell transplant? Did UHR patients who underwent ASCT have a better outcome than those with UHR who did not undergo the procedure?

2.      The details of the death reasons of UHR patients should be provided- authors describe that UHR patients are older, had more comorbidities, and have poor performance status. How many deaths in both groups are truly myeloma-related, and how many of those are, in fact, other reasons related (infections, COVID, exacerbations of chronic illnesses- e.g., heart failure)?

3.      It is questionable whether the Kaplan-Meier method should be used for statistical analysis. Competing risk analysis with death due to other causes and MM-related death seems to be more appropriate.

4.      Figure 1 is quite pointless. The authors defined a priori UHR patients as having overall survival <24 months, so it is obvious that the log-rank test will yield a significant result. Despite that, cumulative incidence (Gray's test for competing risk) of death from MM and other-cause deaths should be visualized.

5.      In table 3, there is no clear how the value of 0.846 in multivariate analysis was obtained. As ISS is a nominal variable with three levels, HR with p-values should be reported for both ISS 2 (ISS 2 vs. ISS 1) and ISS 3 (ISS 3 vs. ISS1).

6.      Conclusions should be revised. E.g., the authors state that: "UHR MM patients with organ dysfunction were very fragile and could 212 not tolerate intensive treatment combining several novel drugs". This statement is not supported by the results, as there were no significant results in induction regimens administered between groups (Table 1).

7.      Potentially more sophisticated statistical methods will ad some merit to the manuscript, e.g., ROC curve for obtained multivariate logistic regression model or nomogram.

8.      The discussion should be elongated, and the number of references should be increased. 

Reviewer 3 Report

Evaluation on patients with multiple myeloma characterized by short survival.

However, the study shows that the most compromised patients have a higher risk of death, as was expected

 Some comments:

-Lines 36-38

I suggest editing the entire paragraph because it’s unclear. Furthermore, a definition of high risk multiple myeloma should be added in the introduction before talking about ultrahigh risk

line 43

I suggest adding biological or disease characteristics

Lines 122-124 and 125

I suggest eliminating the world “that”

Line 137

A reference is required

Lines 191-192

Was light chain Amyloidosis diagnosis excluded?

Thank you